# GENERALIZED ADAPTIVE MOMENT ESTIMATION

## ABSTRACT

Adaptive gradient methods have experienced great success in training deep neural networks (DNNs). The basic idea of the methods is to track and properly make use of the first and/or second moments of the gradient for model-parameter updates over iterations for the purpose of removing the need for manual interference. In this work, we propose a new adaptive gradient method, referred to as *generalized adaptive moment estimation* (Game). From a high level perspective, the new method introduces two more parameters w.r.t. AMSGrad (S. J. Reddi & Kumar (2018)) and one more parameter w.r.t. PAdam (Chen & Gu (2018)) to enlarge the parameter-selection space for performance enhancement while reducing the memory cost per iteration compared to AMSGrad and PAdam. The saved memory space amounts to the number of model parameters, which is significant for large-scale DNNs. Our motivation for introducing additional parameters in Game is to provide algorithmic flexibility to facilitate a reduction of the performance gap between training and validation datasets when training a DNN. Convergence analysis is provided for applying Game to solve both convex optimization and smooth nonconvex optimization. Empirical studies for training four convolutional neural networks over MNIST and CIFAR10 show that under proper parameter selection, Game produces promising validation performance as compared to AMSGrad and PAdam.

## 1 INTRODUCTION

Stochastic gradient descent (SGD) and its variants have become the mainstream training methods in machine learning (ML) due to its simplicity and effectiveness. In general, SGD is known to work reasonably well regardless of their problem structure if the learning rate is set properly in a dynamical manner over training iterations. Intuitively speaking, if optimization problems admit certain structural properties (e.g., gradients magnitudes not balanced across the parameter set), advanced gradient descent methods exploiting the structural properties would be likely to boost optimization performance. In 2011, Duchi et al. firstly proposed to track the second moment of gradients and then scale each gradient coordinate using the tracked information before updating the model parameters, which is referred to as AdaGrad (Duchi et al. (2011)). From a conceptual point of view, AdaGrad adaptively adjusts "individual learning rates" for all the model parameters, allowing for the parameters to be updated on a roughly equal scale. It is found that AdaGrad converges significantly faster than SGD when the gradients are sparse. Its performance deteriorates when the gradients are dense due to a rapid decay of the learning rates.

Since the pioneering work of AdaGrad, various adaptive gradient methods have been proposed on tracking and using the first and/or second moment of the gradients for effective parameter updates. The methods include, for example, RMSProp (Tieleman & Hinton (2012)), Adam (Kingma & Ba (2017)), and NAdam (Dozat (2016)). The main difference of the above methods from AdaGrad is that the first and/or second moment of the gradients are tracked via exponential moving averaging to enhance the importance of the most recent gradients.

Despite the wide usage of Adam in ML community, Reddit et al. have recently shown that the method does not even converge for a class of specially constructed convex functions (S. J. Reddi & Kumar (2018)) due to the non-monotonicity of the "individual learning rates" being multiplied to the gradients. To fix the convergence issue of Adam, the authors proposed a so-called *AMSGrad* method by additionally tracking the maximum value of the second moment of gradients over iterations (i.e., vector $\hat{v}$ of Alg. 1 in Table 1). Later on, Chen and Guo generalized AMSGrad by introducing one free parameter (i.e., $p$ of Alg. 2 in Table 1), referred to as PAdam (Chen & Gu (2018)). The authors'

Table 1: Comparison of three adaptive gradient methods

| Alg. 1: AMSGrad | Alg. 2: PAdam | Alg. 3: Game |
|---|---|---|
| **Input**: $\boldsymbol{x}_1, \{\alpha_t\}, \{\beta_{1t}\}, \beta_2$ | **Input**: $\boldsymbol{x}_1, \{\alpha_t\}, \{\beta_{1t}\}, \beta_2$ | **Input**: $\boldsymbol{x}_1, \{\alpha_t\}, \{\beta_{1t}\}, \beta_2$ |
| **Init.**: $\boldsymbol{m}_0 \leftarrow 0, \hat{\boldsymbol{v}}_0 \leftarrow 0, \boldsymbol{v}_0 \leftarrow 0$ | **Init.**: $\boldsymbol{m}_0 \leftarrow 0, \hat{\boldsymbol{v}}_0 \leftarrow 0, \boldsymbol{v}_0 \leftarrow 0$ | **Init.**: $\boldsymbol{m}_0 \leftarrow 0, \hat{\boldsymbol{v}}_0 \leftarrow 0$ |
| **for** $t = 1$ to $T$ **do** | **for** $t = 1$ to $T$ **do** | **for** $t = 1$ to $T$ **do** |
| $\quad \boldsymbol{g}_t = \nabla_{\boldsymbol{x}} f(\boldsymbol{x}_t; \xi_t)$ | $\quad \boldsymbol{g}_t = \nabla_{\boldsymbol{x}} f(\boldsymbol{x}_t; \xi_t)$ | $\quad \boldsymbol{g}_t = \nabla_{\boldsymbol{x}} f(\boldsymbol{x}_t; \xi_t)$ |
| $\quad \boldsymbol{m}_t = \beta_{1t}\boldsymbol{m}_{t-1} + (1 - \beta_{1t})\boldsymbol{g}_t$ | $\quad \boldsymbol{m}_t = \beta_{1t}\boldsymbol{m}_{t-1} + (1 - \beta_{1t})\boldsymbol{g}_t$ | $\quad \boldsymbol{m}_t = \beta_{1t}\boldsymbol{m}_{t-1} + (1 - \beta_{1t})\boldsymbol{g}_t$ |
| $\quad \boldsymbol{v}_t = \beta_2\boldsymbol{v}_{t-1} + (1 - \beta_2)\boldsymbol{g}_t^2$ | $\quad \boldsymbol{v}_t = \beta_2\boldsymbol{v}_{t-1} + (1 - \beta_2)\boldsymbol{g}_t^2$ | $\quad \boldsymbol{v}_t = \beta_2\hat{\boldsymbol{v}}_{t-1} + (1 - \beta_2)|\boldsymbol{g}_t|^q$ |
| $\quad \hat{\boldsymbol{v}}_t = \max(\boldsymbol{v}_t, \hat{\boldsymbol{v}}_{t-1})$ | $\quad \hat{\boldsymbol{v}}_t = \max(\boldsymbol{v}_t, \hat{\boldsymbol{v}}_{t-1})$ | $\quad \hat{\boldsymbol{v}}_t = \max(\boldsymbol{v}_t, \hat{\boldsymbol{v}}_{t-1})$ |
| $\quad \boldsymbol{x}_{t+1} \leftarrow \boldsymbol{x}_t - \alpha_t \hat{\boldsymbol{V}}_t^{-1/2}\boldsymbol{m}_t$ | $\quad \boldsymbol{x}_{t+1} \leftarrow \boldsymbol{x}_t - \alpha_t \hat{\boldsymbol{V}}_t^{-p}\boldsymbol{m}_t$ | $\quad \boldsymbol{x}_{t+1} \leftarrow \boldsymbol{x}_t - \alpha_t \hat{\boldsymbol{V}}_t^{-p}\boldsymbol{m}_t$ |
| **end for** | **end for** | **end for** |
| Comparison of analysis results for smooth nonconvex optimization | | |
| | (Zhou et al. (2018)) : $O\left(\frac{d}{T} + \frac{G_T}{\sqrt{T}}\right); \beta_1 < \beta_2^{\frac{1}{2}}, p = \frac{1}{4}$ | |
| | new results : $O\left(\frac{d}{T} + \frac{G_T}{\sqrt{T}}\right)$ when $p = 1/4$ | new results : $O\left(\frac{d}{T} + \frac{G_T}{\sqrt{T}}\right)$ when $pq = 1/2$ |
| Notations: $\hat{\boldsymbol{V}}_t = \text{diag}(\hat{\boldsymbol{v}}_t); G_T = \mathbb{E}\left(\sum_{i=1}^d \|\boldsymbol{g}_{1:T,i}\|_2\right); d$ : dimension of $\boldsymbol{x}$ | | |

primary motivation is to improve the generalization performance by tuning the parameter. Both AMSGrad and PAdam require additional memory in comparison to Adam for storing $\hat{\boldsymbol{v}}$.

While machine learning has seen rapid advances in algorithm development, theoretical convergence analysis has also made remarkable progress recently. The work of (Zhou et al. (2018)) extends the original analysis for PAdam in (Chen & Gu (2018)) from convex optimization to smooth nonconvex optimization. In another recent article (Chen et al. (2018)), the authors also considered smooth nonconvex optimization and provided convergence analysis for a class of Adam-related algorithms including AMSGrad and Adam. The convergence bounds developed in the above two articles differ in their analysis approach and step-size selection. From a high level point of view, analysis on nonconvex optimization is highly valuable in practice as training deep neural networks (DNNs) is well known to be a nonconvex optimization problem. An improved convergence analysis on existing algorithms would provide insights on designing more advanced adaptive gradient methods.

In this paper, we make two main contributions. Firstly, we propose a new adaptive gradient method named *generalized adaptive moment estimation* (Game). As shown in Table 1, Game tracks only two variables $(\boldsymbol{m}, \hat{\boldsymbol{v}})$ [1] over iterations as compared to AMSGrad and PAdam, which track three variables $(\boldsymbol{m}, \boldsymbol{v}, \hat{\boldsymbol{v}})$. We emphasize that since the dimension of $\boldsymbol{v}$ is the same as the number of model parameters, the memory saved by Game can be remarkable for modern large-scale neural networks, which usually have millions of parameters. Furthermore, Game introduces two parameters (i.e., $(p, q)$ of Alg. 3 in Table 1) in comparison to PAdam which introduces one parameter (i.e., $p$ of Alg. 2 in Table 1) to further enhance its flexibility. The parameter $q$ enables Game to track information of the $q$th moment of gradient magnitude rather than only the second moment of the gradients. By doing so, the parameter provides one more degree of freedom for the method to balance the tradeoff between convergence speed on training data and generalization ability on validation data.

Secondly, in our theoretical convergence analysis for Game, we manage to remove the condition on the relationship between $\beta_1$ and $\beta_2$ while $\beta_1 \leq \beta_2$ and $\beta_1 \leq \beta_2^p$ are required in (S. J. Reddi & Kumar (2018)) for AMSGrad and (Zhou et al. (2018)) for PAdam, respectively. We provide convergence analysis for both convex optimization and smooth nonconvex optimization. The results for nonconvex case are briefly summarized in Table 1 with the analysis results from (Zhou et al. (2018)) as a reference. Finally, our experimental results on training four convolutional neural networks (CNNs) for MNIST and CIFAR10 suggest that with a proper setup of $(p, q)$, Game produces promising validation performance in comparison to AMSGrad and PAdam.

## 2 PROBLEM SETUP

Before introducing the problem, we describe the notation used in the paper, which are basically in line with that of (S. J. Reddi & Kumar (2018)) and (Chen & Gu (2018)) for consistency. We denote

---

[1] $\boldsymbol{v}$ is not tracked but just computed and used at each iteration.

scalars by lower-case letters, vectors by bold lower-case letters, and matrices by bold upper-case letters. Following convention, we denote the $l_p$ ($p \geq 1$) norm of a vector $\boldsymbol{x} \in \mathbb{R}^d$ by $\|\boldsymbol{x}\|_p = (\sum_{i=1}^d |x_i|^p)^{1/p}$. As $p \to \infty$, the $l_\infty$ norm takes a special form $\|\boldsymbol{x}\|_\infty = \max_{i=1}^d |x_i|$. We let $\boldsymbol{g}_{1:t,i} = [g_{1,i}, g_{2,i}, \cdots, g_{t,i}]$, where $g_{t,i}$ is the $i$th element of a vector $\boldsymbol{g}_t \in \mathbb{R}^d$. We use $\mathbb{E}[\cdot]$ to denote the expectation operation. Given two sequences $\{a_t\}$ and $\{b_t\}$, the notation $a_t = O(b_t)$ indicates that the magnitude of $a_t$ is proportional to that of $b_t$ for $t \geq 0$.

Formally, we reconsider solving the stochastic smooth nonconvex optimization studied in (Ghadimi & Lan (2013; 2016); Zhou et al. (2018))

$$\min_{\boldsymbol{x} \in \mathbb{R}^d} f(\boldsymbol{x}) = \mathbb{E}_\xi [f(\boldsymbol{x}; \xi)], \tag{1}$$

where $\boldsymbol{x}$ represents the parameters of a model in a vector form and $\xi$ is a random variable. Due to randomness of $\xi$, one can only obtain an unbiased noisy gradient $\nabla f(\boldsymbol{x}; \xi)$, satisfying $\nabla f(\boldsymbol{x}) = \mathbb{E}_\xi[\nabla f(\boldsymbol{x}; \xi)]$. In practice, the variable $\xi$ often represents the random mini-batch state. For the above situation, we denote the function realization at iteration $t$ as $f_t(\boldsymbol{x}; \xi_t)$, where $\xi_t$ is a realization of $\xi$. Solving (1) is asymptotically equivalent to addressing the following optimization

$$\lim_{T \to \infty} \min_{\boldsymbol{x} \in \mathbb{R}^d} \sum_{t=1}^T f_t(\boldsymbol{x}; \xi_t). \tag{2}$$

We will consider both the two formulations (1) and (2) in the reminder of the paper.

## 3 GENERALIZED ADAPTIVE MOMENT ESTIMATION (GAME)

In this section, we present our new adaptive gradient method for solving (1). Our algorithm *Game* is motivated by making two observations about AMSGrad and PAdam in Table 1. Firstly, both the two methods have to track the parameter $\hat{\boldsymbol{v}}$ in addition to $\boldsymbol{v}$. In particular, the parameter $\hat{\boldsymbol{v}}_t$ at iteration $t$ is computed as

$$\boldsymbol{v}_t = \beta_2 \boldsymbol{v}_{t-1} + (1 - \beta_2) \boldsymbol{g}_t^2 \tag{3}$$
$$\hat{\boldsymbol{v}}_t = \max(\boldsymbol{v}_t, \hat{\boldsymbol{v}}_{t-1}). \tag{4}$$

It is seen from (3)-(4) that $\boldsymbol{v}_t$ is a function of the squared gradients $\{\boldsymbol{g}_j^2\}_{j=1}^t$ while $\hat{\boldsymbol{v}}_t$ tracks the maximum values of $\{\boldsymbol{v}_j\}_{j=1}^t$ up to iteration $t$. From a high level perspective, it feels redundant to keep both $\boldsymbol{v}_t$ and $\hat{\boldsymbol{v}}_t$ in AMSGrad and PAdam as both parameters are related to the second moment of the gradients. One natural question is if it is sufficient to keep only one parameter about the second moment of the gradients without sacrificing convergence speed. Less memory usage is always desirable in designing an algorithm for simplicity and applicability.

Secondly, it is clear from Table 1 that PAdam is designed by replacing the diagonal matrix $\hat{V}_t^{-\frac{1}{2}}$ with $\hat{V}_t^{-p}$ at iteration $t$ in AMSGrad, where $p \in [0, \frac{1}{2}]$. One can easily show that $p = 0$ corresponds to SGD while $p = \frac{1}{2}$ leads to AMSGrad. The parameter $p$ establishes a smooth connection between SGD and AMSGrad, allowing the resulting method PAdam to carry the advantages of both methods. The empirical study in (Chen & Gu (2018)) suggests that when $p$ is properly chosen in the range $[0, \frac{1}{2}]$, PAdam shows better generalization performance than AMSGrad, which is as expected due to the fact SGD usually produces good generalization performance. To summarize, the above technique of introducing an additional parameter to an existing algorithm adds one more degree of freedom to enhance its applicability. One can find many examples in applied mathematics that use similar techniques for knowledge expansion. For instance, the extension from $l_2$ norm to $l_p$ norm and generalization from Cauchy-Schwarz inequality to Hölder's inequality.

Based on the above two observations, we propose a new adaptive gradient method named *Game* as summarized in Table 1. Basically, the new method only tracks two parameters $(\boldsymbol{m}, \hat{\boldsymbol{v}})$, where $\hat{\boldsymbol{v}}_t$ at iteration $t$ is computed as

$$\boldsymbol{v}_t = \beta_2 \hat{\boldsymbol{v}}_{t-1} + (1 - \beta_2) |\boldsymbol{g}_t|^q \tag{5}$$
$$\hat{\boldsymbol{v}}_t = \max(\boldsymbol{v}_t, \hat{\boldsymbol{v}}_{t-1}), \tag{6}$$

where $q > 0$ is our newly introduced parameter and $|\boldsymbol{g}_t|^q$ denotes element-wise $q$th power of $|\boldsymbol{g}_t|$. It is clear from (5)-(6) that $\boldsymbol{v}_t$ is only a temporary parameter at iteration $t$ and is not required to be stored

in the memory for next iteration. Therefore, it is safe to say that Game saves roughly 33 percent memory per iteration as compared to AMSGrad and PAdam by removing the need for tracking $\{\boldsymbol{v}_t\}$.

We now study the impact of saved memory by Game when training large-scale neural networks. Note that the dimension of $\boldsymbol{v}$ is actually the number of model parameters. That is, Game manages to save a memory space which is equivalent to the size of the neural network to be trained. It is known that state-of-the-art neural networks for challenging tasks (e.g., ImageNet competition (Russakovsky et al. (2015))) tend to be extremely deep and consist of millions of parameters (Simonyan & Zisserman (2016)). When applying Game to train those large-scale networks, the memory saved by the new training method becomes remarkable with AMSGrad as a reference.

Next we let $q = 2$ and $p = \frac{1}{2}$ for Game to make a fair comparison w.r.t. AMSGrad . In this case, one can easily show that $\hat{v}_t$ in (6) is either greater than or equal to the one in (4). As a result, Game would always have either equal or smaller effective learning rate $\alpha_t \hat{\boldsymbol{V}}_t^{-\frac{1}{2}}$ than AMSGrad. The above property implies that Game is more conservative than AMSGrad and PAdam due to the update reformulation (5)-(6). If needed, one can adjust the parameters $\{\alpha_t\}$ to larger values to make Game more aggressive.

Finally we consider the new scalar parameter $q$ introduced in (5). From an algebraic point of view, introduction of parameter $q$ allows $\hat{\boldsymbol{v}}$ in (5)-(6) to track information of the $q$th moment of the gradient magnitude over iterations. As $q$ decreases, small gradients would be amplified while large gradients would be suppressed, leading to a decreasing dynamic range of $\hat{\boldsymbol{v}}$. When $q \to 0$, it is not difficult to show that Game approaches SGD. Therefore, the parameter $q$ has a similar effect as the parameter $p$ of PAdam. We point out that when $\beta_2 > 0$, we cannot merge $p$ and $q$ into one parameter via certain reformulation of the updating expressions of Game. That is, $p$ and $q$ play different roles in Game. Our purpose of introducing $q$ in addition to $p$ is to enlarge the parameter-selection space of Game and improve generalization performance with proper parameter setup.

## 4 CONVERGENCE ANALYSIS FOR GAME

In this section, we analyze the convergence of Game for both convex optimization and smooth nonconvex optimization. The cost regret will be studied for the convex case while gradient expectation will be considered for the nonconvex case, which represents two different analysis approaches.

### 4.1 ANALYSIS FOR CONVEX OPTIMIZATION

Formally, we define the cost regret up to iteration $T$ as

$$R_T = \sum_{t=1}^{T} \left[ f_t(\boldsymbol{x}_t; \xi_t) - f_t(\boldsymbol{x}_T^*; \xi_t) \right], \tag{7}$$

where $\boldsymbol{x}_T^* = \arg\min \sum_{t=1}^{T} f_t(\boldsymbol{x})$ denotes the optimal solution within the iteration range $[1, T]$, and $\boldsymbol{x}_t$ is a causal estimate of $\boldsymbol{x}_T^*$ at iteration $t$ as obtained by following Game in Table 1. Our objective is to derive an upper bound of $R_T$ and then quantify its convergence behaviour as $T$ increases.

Next we present our convergence analysis:

**Lemma 1.** *Let* $(\beta_1, \beta_2) \in [0, 1)$, $\beta_{1t} \leq \beta_1$, $pq < 2$ *and* $p, q > 0$, *and* $\alpha_t = \alpha/\sqrt{t}$. *Then the quantity* $\sum_{t=1}^{T} \alpha_t \left[ \|\hat{\boldsymbol{V}}_t^{-p/2} \boldsymbol{m}_t\|_2^2 \right]$ *of Game is upper bounded by*

$$\sum_{t=1}^{T} \alpha_t \left[ \|\hat{\boldsymbol{V}}_t^{-p/2} \boldsymbol{m}_t\|_2^2 \right] \leq \frac{\alpha\sqrt{1 + \log T}}{(1 - \beta_1)^2 (1 - \beta_2)^p} \left[ \sum_{i=1}^{d} \left( \sum_{j=1}^{T} |g_{j,i}|^{2(2-pq)} \right)^{1/2} \right]. \tag{8}$$

*Proof.* See Appendix A for proof. □

**Remark 1.** *We emphasize that there is a major difference between the upper bound expression in (8) and those derived in (S. J. Reddi & Kumar (2018)) and (Chen & Gu (2018)) for analyzing AMSGrad and PAdam. That is the new upper bound does not put any restriction on the relationship between $\beta_1$ and $\beta_2$ for (8) to hold while in the above two articles, it is required that $\beta_1 \leq \beta_2^{2p}$, where $p$ is the parameter of PAdam.*

**Theorem 1** (convex). *Suppose $\{f_t\}$ are close, proper and convex functions (Sawaragi et al. (1985)). Let $\beta_1, \beta_2 \in [0, 1)$, $\beta_{1t} \leq \beta_1$, $pq < 2$ and $p, q > 0$, and $\alpha_t = \alpha/\sqrt{t}$. Assume $\|\nabla f_t(\boldsymbol{x}; \xi_t)\|_\infty \leq G_\infty$ for all $t \leq T$ and distance between any $\boldsymbol{x}_t$ generated by Game and $\boldsymbol{x}_T^*$ is bounded, i.e., $\|\boldsymbol{x}_m - \boldsymbol{x}_T^*\|_\infty \leq D_\infty$. Then we have the following bound on the regret*

$$
\begin{aligned}
R_T \leq & \frac{D_\infty^2 \sqrt{T}}{\alpha(1 - \beta_1)} \sum_{i=1}^d \hat{v}_{T,i}^p + \frac{D_\infty^2}{2(1 - \beta_1)} \sum_{t=1}^T \sum_{i=1}^d \frac{\beta_{1t} \hat{v}_{t,i}^p}{\alpha_t} \\
& + \frac{\alpha \sqrt{1 + \log T}}{(1 - \beta_1)^3 (1 - \beta_2)^p} \left[ \sum_{i=1}^d \left( \sum_{j=1}^T |g_{j,i}|^{2(2-pq)} \right)^{1/2} \right]
\end{aligned}
\tag{9}
$$

Based on Lemma 1, one can easily derive the upper bound expression (9) by following the derivation steps for Theorem 4 in (S. J. Reddi & Kumar (2018)). When $pq = 1$ and $p = 1/2$, (9) can be simplified as

$$
R_T \leq \frac{D_\infty^2 \sqrt{T}}{\alpha(1 - \beta_1)} \sum_{i=1}^d \hat{v}_{T,i}^{1/2} + \frac{D_\infty^2}{2(1 - \beta_1)} \sum_{t=1}^T \sum_{i=1}^d \frac{\beta_{1t} \hat{v}_{t,i}^p}{\alpha_t} + \frac{\alpha \sqrt{1 + \log T}}{(1 - \beta_1)^3 (\sqrt{1 - \beta_2}} \left[ \sum_{i=1}^d \|\boldsymbol{g}_{1:T,i}\|_2 \right]. \tag{10}
$$

One can see that no constraint is imposed between the relationship of $\beta_1$ and $\beta_2$ for the upper bound to hold. We have conducted empirical studies and found that both AMSGrad and Game converge even when $\beta_2 < \beta_1$.

## 4.2 ANALYSIS FOR SMOOTH NONCONVEX OPTIMIZATION

Differently from the analysis for convex optimization, we will consider gradient expectation for nonconvex case. To simplify study later on, we choose the output $\boldsymbol{x}_{out}$ from $\{\boldsymbol{x}_t\}_{t=2}^T$ with probability $\alpha_{t-1}/(\sum_{j=1}^{T-1} \alpha_j)$, which is in line with the definition in (Zhou et al. (2018)). In practice, it is natural to take the most recent estimate $\boldsymbol{x}_T$ at last iteration $T$ as output.

We now introduce the $L$-smooth assumption needed for analysis:

**Assumption 1** ($L$-smooth). *$f(\boldsymbol{x}) = \mathbb{E}_\xi f(\boldsymbol{x}; \xi)$ is $L$-smooth: for any $\boldsymbol{x}, \boldsymbol{y} \in \mathbb{R}^d$, we have*

$$
|f(\boldsymbol{x}) - f(\boldsymbol{y}) + \langle \nabla f(\boldsymbol{y}), \boldsymbol{x} - \boldsymbol{y} \rangle| \leq \frac{L}{2} \|\boldsymbol{x} - \boldsymbol{y}\|_2^2. \tag{11}
$$

*Furthermore, $f(\boldsymbol{x})$ is lower bounded, i.e., $\inf_{\boldsymbol{x}} f(\boldsymbol{x}) > -\infty$.*

The above smooth assumption is standard for nonconvex optimization. It essentially requires the object function $f$ changes smoothly in the parameter space. See (Zhou et al. (2018); Chen et al. (2018)) for employing the assumption in their analysis.

We now present our convergence analysis in two steps. Firstly, we provide upper bounds for the two quantities $\sum_{t=1}^T \alpha_t^2 \mathbb{E}\left[\|\hat{\boldsymbol{V}}_t^{-p} \boldsymbol{m}_t\|_2^2\right]$ and $\sum_{t=1}^T \alpha_t^2 \mathbb{E}\left[\|\hat{\boldsymbol{V}}_t^{-p} \boldsymbol{g}_t\|_2^2\right]$ in a lemma below. We then show the main result in a theorem, which are derived based on the lemma.

**Lemma 2.** *Let $\beta_1, \beta_2 \in [0, 1)$, $pq \leq 1$ and $p, q > 0$, the step sizes $\alpha_j \leq \alpha_{j-1}$ for all $j > 1$, and $r \in [2pq - 1, 1]$. Assuming $\|\nabla f_t(\boldsymbol{x}; \xi_t)\|_\infty \leq G_\infty$ for all $t \leq T$, we then have*

$$
\sum_{t=1}^T \alpha_t^2 \mathbb{E}\left[\|\hat{\boldsymbol{V}}_t^{-p} \boldsymbol{m}_t\|_2^2\right] \leq \frac{\alpha_1^2 G_\infty^{(1+r-2pq)} T^{(1+r)/2}}{(1 - \beta_2)^{2p}} \mathbb{E}\left[\sum_{i=1}^d (\|\boldsymbol{g}_{1:T,i}\|_2)^{(1-r)}\right] \tag{12}
$$

$$
\sum_{t=1}^T \alpha_t^2 \mathbb{E}\left[\|\hat{\boldsymbol{V}}_t^{-p} \boldsymbol{g}_t\|_2^2\right] \leq \frac{\alpha_1^2 T^{pq}}{(1 - \beta_2)^{2p}} \mathbb{E}\left[\sum_{i=1}^d (\|\boldsymbol{g}_{1:T,i}\|_2)^{2(1-pq)}\right]. \tag{13}
$$

*Proof.* See Appendix B for proof. □

We note that a scalar parameter $r \in [2pq - 1, 1]$ is introduced in the upper bound expression of Lemma 2. As will be discussed in Corollary 1 later on, the parameter $r$ provides a freedom to tune the expression (12) and merge with other expressions for simplicity.

**Remark 2.** *Again the two upper bounds (12)-(13) do not require any constraint on the relationship of $\beta_1$ and $\beta_2$, which is consistent with the results in Lemma 1.*

**Theorem 2** (nonconvex). *Let $\beta_1, \beta_2 \in [0, 1)$, $pq < 1$ and $p, q > 0$, the step sizes $\alpha_t = \alpha_1$ for all $t > 1$, and $r \in [2pq - 1, 1]$. Assume $\|\nabla f_t(\boldsymbol{x}; \xi_t)\|_\infty \leq G_\infty$ for all $t \leq T$ and Assumption 1 holds. The output $\boldsymbol{x}_{out}$ of Game satisfies*

$$\mathbb{E}\left[\|\nabla f(\boldsymbol{x}_{out})\|_2^2\right] \leq \frac{M_1}{T-1} + \frac{M_2 d}{T-1} + \frac{M_3 T^{pq}}{T-1} \mathbb{E}\left(\sum_{i=1}^{d} (\|\boldsymbol{g}_{1:T,i}\|_2)^{2(1-pq)}\right)$$

$$+ \frac{M_4 T^{(1+r)/2}}{T-1} \mathbb{E}\left(\sum_{i=1}^{d} (\|\boldsymbol{g}_{1:T,i}\|_2)^{1-r}\right) \quad T \geq 2, \tag{14}$$

*where*

$$M_1 = G_\infty^{pq} \Delta f / \alpha_1 \tag{15}$$

$$M_2 = \frac{G_\infty^{2+pq} \|\hat{\boldsymbol{v}}_1^{-p}\|_1}{d(1 - \beta_1)} + \frac{G_\infty^2}{(1 - \beta_2)^p} \tag{16}$$

$$M_3 = \frac{2LG_\infty^{pq}\alpha_1}{(1 - \beta_2)^{2p}} \tag{17}$$

$$M_4 = \frac{4LG_\infty^{1+r-pq}\alpha_1}{(1 - \beta_2)^{2p}} \left(\frac{\beta_1}{1 - \beta_1}\right)^2 \tag{18}$$

*where $\Delta f = f(\boldsymbol{x}_1) - \inf_{\boldsymbol{x}} f(\boldsymbol{x})$.*

*Proof.* See Appendix C for proof. ☐

It is clear from Theorem 2 that the two parameters $p$ and $q$ have to satisfy certain conditions. That is $p, q > 0$ and $pq < 1$. The conditions suggest if one parameter is chosen large, the other one should be set small to avoid divergence. In practice, it is found that Game also converges when $pq = 1$. This leaves us an open question on how to elaborate the convergence analysis to cover the special case of $pq = 1$ for Game.

We now study the upper bound on the right hand side of (14). The expression includes four quantities, where the first two quantities are independent of the gradients $\{\boldsymbol{g}_{1:T,i} | i = 1, \ldots, d\}$ while the last two are contributed by the inequalities derived in Lemma 2. All the four scalar parameters $M_i$, $i = 1, \ldots, 4$, are independent of iteration number $T$. The convergence of Game can be established if one can show that the upper bound expression approaches to zero as $T$ increases.

In the following, we simplify the upper bound expression in (14) by specifying $pq$ and $r$ with particular values and then study its convergence behaviour.

**Corollary 1.** *Let $pq = \frac{1}{2}$ and $r = 2pq - 1$ in Theorem 2, the upper bound of $\mathbb{E}\left[\|\nabla f(\boldsymbol{x}_{out})\|_2^2\right]$ then takes the form of*

$$\mathbb{E}\left[\|\nabla f(\boldsymbol{x}_{out})\|_2^2\right] \leq \frac{M_1}{(T-1)\alpha_1} + \frac{M_2 d}{T-1} + \frac{(M_3 + M_4)\alpha_1\sqrt{T}}{T-1} \mathbb{E}\left(\sum_{i=1}^{d} \|\boldsymbol{g}_{1:T,i}\|_2\right) \quad T \geq 2. \tag{19}$$

As summarized in Table 1, It is immediate that the upper bound in (19) is proportional to $O(d/T + G_T/\sqrt{T})$, where $G_T = \mathbb{E}\left(\sum_{i=1}^{d} \|\boldsymbol{g}_{1:T,i}\|_2\right)$. As reflected in Table 1, the upper bound also holds for PAdam even though the iteration procedure (3)-(4) of PAdam is different from (5)-(6) of Game.

We are now in a position to study under what conditions Game would converge. It is not difficult to conclude that when $G_T$ is of order $G_T = O((dT)^s)$ where $s < \frac{1}{2}$, (19) tends to converge with

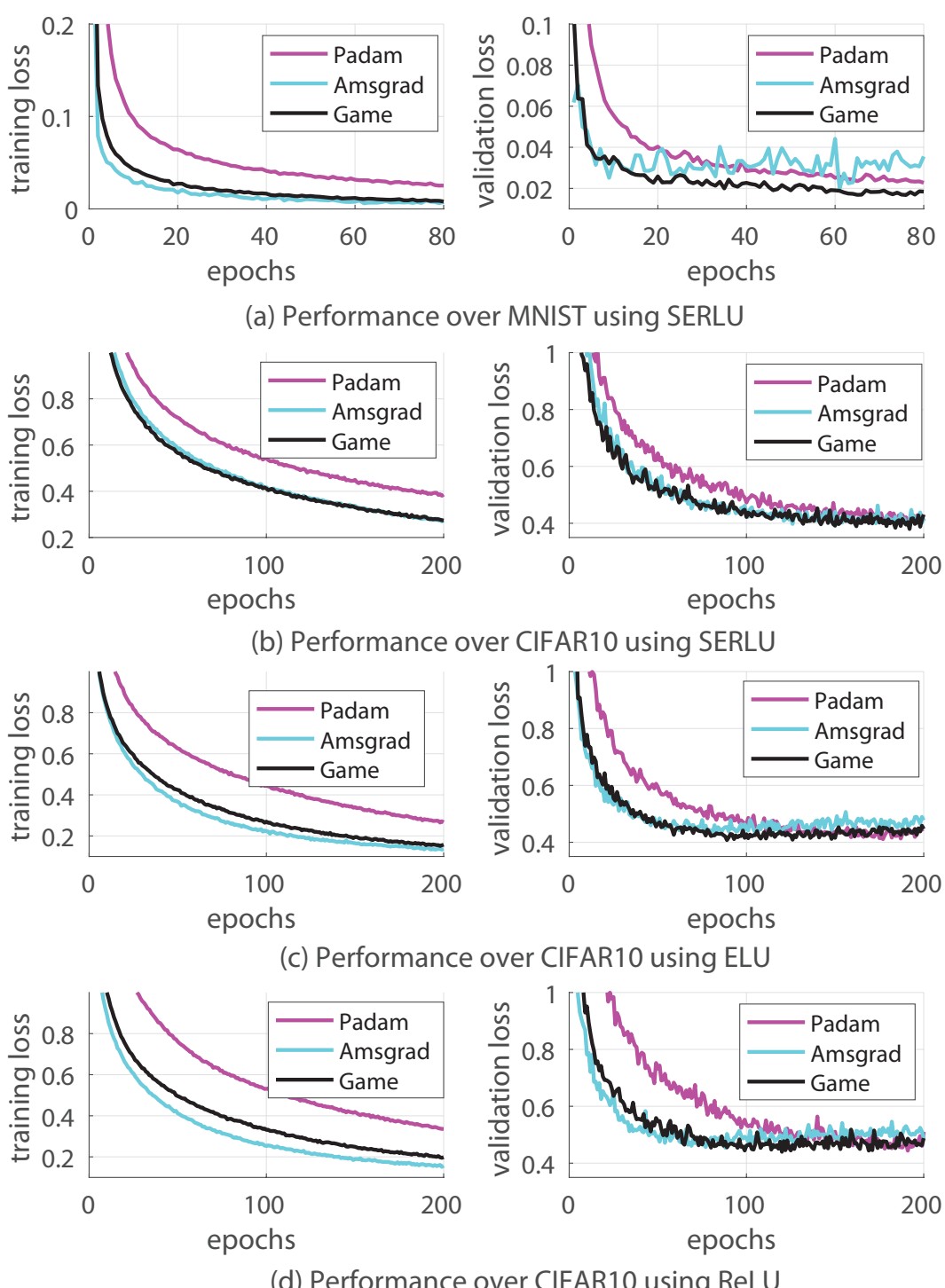

Figure 1: Performance comparison of PAdam, AMSGrad, and Game for training four CNNs (see Table 2-5 for the detailed CNN architectures in Appendix D).

the speed $O(d/T + d^s/T^{1/2-s})$. We note that in Duchi et al. (2011)), the assumption $G_T \ll \sqrt{dT}$ was used for analyzing AdaGrad, which was later on employed for convergence analysis of other adaptive gradient methods (see (Zhou et al. (2018)) and (S. J. Reddi & Kumar (2018)) for example). The assumption $G_T = O((dT)^s)$ is slightly stronger than $G_T \ll \sqrt{dT}$ as in practice, the dimension $d$ could be larger than $T$ when training large-scale neural networks.

## 5 EXPERIMENTAL RESULTS

In the experiment, we evaluated the effectiveness of AMSGrad, PAdam and Game for training convolutional neural networks (CNNs). The classification problems on the two datasets MNIST and CIFAR10 were considered. To alleviate overfitting, each of the two datasets was augmented with additional training data (e.g., shifting images vertically and/or horizontally, and image flipping for CIFAR10). In brief, we tested four CNNs: the 1st one consists of four layers for MNIST using the activation function SERLU (Zhang & Li (2018)), the 2nd consists of 8 layers for CIFAR10 using SERLU, and the 3rd and 4th have the same structure as the 2nd one but using ELU (Clevert et al. (2016)) and ReLU (Nair & Hinton (2010)), respectively. More detailed information of the four CNNs can be found in Table 2-5 of Appendix D.

We first consider the CNN training for MNIST using SERLU. The shared parameters among the three methods include $(\alpha_t, \beta_1, \beta_2) = (0.001, 0.9, 0.999)$, which was recommended in (Kingma & Ba (2017)) for Adam. On the other hand, the special parameters include $p = 0.125$ for PAdam[2] and $(p, q) = (0.5, 1)$ for Game. The setup $(p, q) = (0.5, 1)$ was found empirically to be more effective than other tested values. It is worth noting that $q = 1$ indicates that Game tracks information of the gradient magnitude rather than the second moment of gradients.

Next we study the CNN training for CIFAR10 using SERLU, ELU and ReLU. As the three neural networks (see Table 3-5) are deeper than that for MNIST, we employed shift-dropout for SERLU and dropout for ELU and ReLU to combat overfitting, respectively. The parameter setup is slightly different from the one for MNIST. In particular, the shared parameters among the three methods include $(\beta_1, \beta_2) = (0.9, 0.999)$ and dropout rate of 0.2. Special parameters include $\alpha_t = 0.0001$ for AMSGrad, [3] $(\alpha_t, p) = (0.001, 0.125)$ for PAdam, and $(\alpha_t, p, q) = (0.001, 0.5, 1)$ for Game.

The convergence results for training the four CNNs are demonstrated in Figure 1. It is seen that Game produces better generalization performance (on validation datasets) than AMSGrad in subplot $(a)$ and $(c) - (d)$ while in the same three subplots, AMSGrad exhibits the fastest convergence speed over training data. The two methods have roughly the same convergence performance in subplot $(b)$, suggesting that selection of activation functions also affects the convergence behaviours. It is observed that SERLU performs slightly better than ELU and ReLU in terms of validation loss. In all the four subplots, PAdam converges slightly slower due to the fact the a small value $p = 0.125$ is selected as compared to $p = 0.5$ of AMSGrad.

To briefly summarize, the above convergence results suggest that Game produces promising generalization performance in comparison to AMSGrad and PAdam on validation datasets. The nice convergence behaviour of Game might be because the method tracks the 1st moment of gradient magnitude by setting $q = 1$. As a result, Game achieves better validation performance by slightly sacrificing convergence speed on training dataset. If one also takes into account of memory usage (see Table 1), it is clear that Game is simpler to implement and requires less memory resource, rendering Game an advantage over AMSGrad and PAdam.

## 6 CONCLUSIONS AND FUTURE WORKS

In this paper, we have proposed a new adaptive gradient method termed as Game. The new method only needs to track two parameters $(\boldsymbol{m}, \hat{\boldsymbol{v}})$ in comparison to AMSGrad and PAdam, which track three parameters $(\boldsymbol{m}, \boldsymbol{v}, \hat{\boldsymbol{v}})$, thus requiring only two-thirds of memory w.r.t. the two methods. The saved memory scales along with the number of model parameters, which becomes significant when training large-scale neural networks. Furthermore, Game introduces one additional parameter $q$, allowing the method to track information of the $q$th moment of the gradient magnitude, while AMSGrad and PAdam only consider tracking the 2nd moment of gradients. The freedom of tuning parameter $q$ makes Game more flexible. Theoretical convergence analysis is provided for applying Game to solve both convex and smooth nonconvex optimization. Experimental results on MNIST and CIFAR10 demonstrate that Game produces promising generalization performance in comparison to AMSGrad and PAdam. Given that Game demands only two-thirds of memory in implementation, we conclude that the new method is a promising candidate for training large-scale neural networks.

---

[2]This parameter value $p = 0.125$ was suggested by the authors of (Chen & Gu (2018)).

[3]It is found that $\alpha_t = 0.001$ is not stable when applying AMSGrad to train the CNNs in this case.

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

APPENDIX

## A PROOF OF LEMMA 1

Before formally presenting the proof, we first introduce a lemma below:

**Lemma 3.** *Let $\hat{v}_t$ be defined as in Game and $q > 0$. Then each component $\hat{v}_{t,i}$ is lower bounded by $\hat{v}_{t,i} \geq (1 - \beta_2)|g_{t,i}|^q$, $i = 1, \ldots, d$.*

*Proof.* The proof follows directly from the computation $\hat{v}_{t,i} = \max(\hat{v}_{t-1,i}, \beta_2\hat{v}_{t-1,i} + (1-\beta_2)|g_{t,i}|^q)$ as summarized in Table 1 for Game. $\square$

With Lemma 3, we are ready to describe the proof for Lemma 1. Firstly, the quantity $\alpha_t\mathbb{E}\left[\|\hat{V}_t^{-p/2}m_t\|_2^2\right]$ can be upper bounded by

$$\alpha_t\left[\|\hat{V}_t^{-p/2}m_t\|_2^2\right]$$

$$= \alpha_t\left[\sum_{i=1}^d \frac{m_{t,i}^2}{\hat{v}_{t,i}^p}\right]$$

$$= \alpha_t\left[\sum_{i=1}^d \frac{1}{\hat{v}_{t,i}^p}\left(\sum_{j=1}^t (1-\beta_{1j})\prod_{k=1}^{t-j}\beta_{1(t-k+1)}g_{j,i}\right)^2\right]$$

$$\overset{(a)}{\leq} \alpha_t\left[\sum_{i=1}^d \frac{1}{\hat{v}_{t,i}^p}\left(\sum_{j=1}^t \beta_1^{t-j}g_{j,i}\right)^2\right]$$

$$\overset{(b)}{\leq} \alpha_t\left[\sum_{i=1}^d \frac{1}{\hat{v}_{t,i}^p}\left(\sum_{j=1}^t \beta_1^{t-j}\right)\left(\sum_{j=1}^t \beta_1^{t-j}|g_{j,i}|^2\right)\right]$$

$$\overset{(c)}{\leq} \frac{\alpha_t}{1-\beta_1}\left[\sum_{i=1}^d \frac{1}{\hat{v}_{t,i}^p}\left(\sum_{j=1}^t \beta_1^{t-j}|g_{j,i}|^2\right)\right]$$

$$\overset{(d)}{\leq} \frac{\alpha_t}{1-\beta_1}\left[\sum_{i=1}^d \sum_{j=1}^t \frac{1}{\hat{v}_{j,i}^p}\beta_1^{t-j}|g_{j,i}|^2\right]$$

$$\overset{(e)}{\leq} \frac{\alpha_t}{(1-\beta_1)(1-\beta_2)^p}\left[\sum_{i=1}^d \sum_{j=1}^t \beta_1^{t-j}|g_{j,i}|^{(2-pq)}\right] \tag{20}$$

where step $(a)$ uses $\beta_{1t} \leq \beta_1 < 1$, step $(b)$ uses Cauchy-Schwarz inequality, step $(c)$ uses $\sum_{j=1}^t \beta_1^{t-j} \leq 1/(1-\beta_1)$, step $(d)$ uses $\hat{v}_{t,i}^p \geq \hat{v}_{j,i}^p$ for all $j \leq t$, and step $(e)$ uses the results in Lemma 3.

Next taking the summation of (20) over $t = 1$ to $t = T$ produces

$$\sum_{t=1}^T \alpha_t\left[\|\hat{V}_t^{-p/2}m_t\|_2^2\right]$$

$$\leq \frac{1}{(1-\beta_1)(1-\beta_2)^p}\left[\sum_{t=1}^T \sum_{i=1}^d \sum_{j=1}^t \alpha_t\beta_1^{t-j}|g_{j,i}|^{(2-pq)}\right]$$

$$\overset{(a)}{\leq} \frac{\alpha}{(1-\beta_1)(1-\beta_2)^p}\left[\sum_{i=1}^d \sum_{j=1}^T |g_{j,i}|^{(2-pq)}\sum_{t=j}^T \frac{\beta_1^{t-j}}{\sqrt{t}}\right]$$

$$\leq \frac{\alpha}{(1-\beta_1)(1-\beta_2)^p}\left[\sum_{i=1}^d \sum_{j=1}^T \frac{1}{\sqrt{j}}|g_{j,i}|^{(2-pq)}\sum_{t=j}^T \beta_1^{t-j}\right]$$

$$\overset{(b)}{\leq} \frac{\alpha}{(1-\beta_1)^2(1-\beta_2)^p} \left[ \sum_{i=1}^{d} \sum_{j=1}^{T} \frac{1}{\sqrt{j}} |g_{j,i}|^{(2-pq)} \right]$$

$$\overset{(c)}{\leq} \frac{\alpha}{(1-\beta_1)^2(1-\beta_2)^p} \left[ \sum_{i=1}^{d} \left( \sum_{j=1}^{T} |g_{j,i}|^{2(2-pq)} \right)^{1/2} \left( \sum_{j=1}^{T} \frac{1}{j} \right)^{1/2} \right]$$

$$\overset{(d)}{\leq} \frac{\alpha\sqrt{1+\log T}}{(1-\beta_1)^2(1-\beta_2)^p} \left[ \sum_{i=1}^{d} \left( \sum_{j=1}^{T} |g_{j,i}|^{2(2-pq)} \right)^{1/2} \right] \tag{21}$$

where step $(a)$ uses $\alpha_t = \alpha/\sqrt{t}$, step $(b)$ uses $\sum_{t=j}^{T} \beta_1^{t-j} < 1/(1-\beta_1)$, step $(c)$ follows from Cauchy-Schwarz inequality, and step $(d)$ uses $\sum_{t=1}^{T} 1/t \leq (1+\log T)$. The proof is complete.

## B  PROOF OF LEMMA 2

We first consider the summation $\sum_{t=1}^{T} \alpha_t^2 \mathbb{E} \left[ \|\hat{V}_t^{-p} m_t\|_2^2 \right]$. Each quantity $\alpha_t^2 \mathbb{E} \left[ \|\hat{V}_t^{-p} m_t\|_2^2 \right]$ can be upper bounded by

$$\alpha_t^2 \mathbb{E} \left[ \|\hat{V}_t^{-p} m_t\|_2^2 \right]$$

$$= \alpha_t^2 \mathbb{E} \left[ \sum_{i=1}^{d} \frac{m_{t,i}^2}{\hat{v}_{t,i}^{2p}} \right]$$

$$= \alpha_t^2 \mathbb{E} \left[ \sum_{i=1}^{d} \frac{1}{\hat{v}_{t,i}^{2p}} \left( \sum_{j=1}^{t} (1-\beta_1)\beta_1^{t-j} g_{j,i} \right)^2 \right]$$

$$\overset{(a)}{\leq} \alpha_t^2 (1-\beta_1)^2 \mathbb{E} \left[ \sum_{i=1}^{d} \frac{1}{\hat{v}_{t,i}^{2p}} \left( \sum_{j=1}^{t} \beta_1^{t-j} |g_{j,i}|^{(1+r-2pq)} \right) \left( \sum_{j=1}^{t} \beta_1^{t-j} |g_{j,i}|^{(1-r+2pq)} \right) \right]$$

$$\overset{(b)}{\leq} \alpha_t^2 (1-\beta_1)^2 \mathbb{E} \left[ \sum_{i=1}^{d} \frac{1}{\hat{v}_{t,i}^{2p}} \left( \sum_{j=1}^{t} \beta_1^{t-j} G_\infty^{(1+r-2pq)} \right) \left( \sum_{j=1}^{t} \beta_1^{t-j} |g_{j,i}|^{(1-r+2pq)} \right) \right]$$

$$\overset{(c)}{\leq} \alpha_t^2 (1-\beta_1) G_\infty^{(1+r-2pq)} \mathbb{E} \left[ \sum_{i=1}^{d} \frac{1}{\hat{v}_{t,i}^{2p}} \left( \sum_{j=1}^{t} \beta_1^{t-j} |g_{j,i}|^{(1-r+2pq)} \right) \right]$$

$$\overset{(d)}{\leq} \alpha_t^2 (1-\beta_1) G_\infty^{(1+r-2pq)} \mathbb{E} \left[ \sum_{i=1}^{d} \sum_{j=1}^{t} \frac{1}{\hat{v}_{j,i}^{2p}} \beta_1^{t-j} |g_{j,i}|^{(1-r+2pq)} \right]$$

$$\overset{(e)}{\leq} \frac{\alpha_t^2 (1-\beta_1) G_\infty^{(1+r-2pq)}}{(1-\beta_2)^{2p}} \mathbb{E} \left[ \sum_{i=1}^{d} \sum_{j=1}^{t} \beta_1^{t-j} |g_{j,i}|^{(1-r)} \right] \tag{22}$$

where step $(a)$ uses Cauchy-Schwarz inequality and the property that $2pq - 1 \leq r \leq 1$ and $pq \leq 1$, step $(b)$ uses $|g_{j,i}| \leq \|g_j\|_\infty \leq G_\infty$, step $(c)$ uses $\sum_{j=1}^{t} \beta_1^{t-j} \leq 1/(1-\beta_1)$, step $(d)$ uses $\hat{v}_{t,i}^{2p} \geq \hat{v}_{j,i}^{2p}$ for all $j \leq t$, and step $(e)$ uses the results in Lemma 3.

Taking the summation of (22) over $t = 1$ to $t = T$ produces

$$\sum_{t=1}^{T} \alpha_t^2 \mathbb{E} \left[ \|\hat{V}_t^{-p} m_t\|_2^2 \right]$$

$$\leq \frac{(1-\beta_1) G_\infty^{(1+r-2pq)}}{(1-\beta_2)^{2p}} \mathbb{E} \left[ \sum_{t=1}^{T} \sum_{i=1}^{d} \sum_{j=1}^{t} \alpha_t^2 \beta_1^{t-j} |g_{j,i}|^{(1-r)} \right] \tag{23}$$

$$\overset{(a)}{\leq} \frac{\alpha_1^2 (1 - \beta_1) G_\infty^{(1+r-2pq)}}{(1 - \beta_2)^{2p}} \mathbb{E} \left[ \sum_{i=1}^{d} \sum_{j=1}^{T} |g_{j,i}|^{(1-r)} \sum_{t=j}^{T} \beta_1^{t-j} \right]$$

$$\overset{(b)}{\leq} \frac{\alpha_1^2 G_\infty^{(1+r-2pq)}}{(1 - \beta_2)^{2p}} \mathbb{E} \left[ \sum_{i=1}^{d} \sum_{j=1}^{T} |g_{j,i}|^{(1-r)} \right]$$

$$\overset{(c)}{\leq} \frac{\alpha_1^2 G_\infty^{(1+r-2pq)}}{(1 - \beta_2)^{2p}} \mathbb{E} \left[ \sum_{i=1}^{d} \left( \sum_{j=1}^{T} |g_{j,i}|^2 \right)^{(1-r)/2} \right] T^{(1+r)/2}$$

$$= \frac{\alpha_1^2 G_\infty^{(1+r-2pq)} T^{(1+r)/2}}{(1 - \beta_2)^{2p}} \mathbb{E} \left[ \sum_{i=1}^{d} \left( \|\boldsymbol{g}_{1:T,i}\|_2 \right)^{(1-r)} \right]$$

where step $(a)$ uses the property that $a_j \leq a_{j-1}$ for all $j > 1$, step (b) uses $\sum_{t=j}^{T} \beta_1^{t-j} < 1/(1 - \beta_1)$, and step $(c)$ follows from Holder's inequality. The proof is complete for (12).

The quantity $\sum_{t=1}^{T} \alpha_t^2 \mathbb{E} \left[ \|\hat{\boldsymbol{V}}_t^{-p} \boldsymbol{g}_t\|_2^2 \right]$ in (13) can be upper bounded as

$$\sum_{t=1}^{T} \alpha_t^2 \mathbb{E} \left[ \|\hat{\boldsymbol{V}}_t^{-p} \boldsymbol{g}_t\|_2^2 \right]$$

$$= \sum_{t=1}^{T} \alpha_t^2 \mathbb{E} \left[ \sum_{i=1}^{d} \frac{g_{t,i}^2}{\hat{v}_{t,i}^{2p}} \right]$$

$$\overset{(a)}{\leq} \sum_{t=1}^{T} \alpha_1^2 \mathbb{E} \left[ \sum_{i=1}^{d} \frac{g_{t,i}^2}{(1 - \beta_2)^{2p} |g_{t,i}|^{2pq}} \right]$$

$$\overset{(b)}{=} \frac{\alpha_1^2}{(1 - \beta_2)^{2p}} \mathbb{E} \left[ \sum_{i=1}^{d} \sum_{j=1}^{T} |g_{j,i}|^{2-2pq} \right]$$

$$\overset{(c)}{\leq} \frac{\alpha_1^2}{(1 - \beta_2)^{2p}} \mathbb{E} \left[ \sum_{i=1}^{d} \left( \sum_{j=1}^{T} |g_{j,i}|^2 \right)^{1-pq} \right] T^{pq}$$

$$= \frac{\alpha_1^2 T^{pq}}{(1 - \beta_2)^{2p}} \mathbb{E} \left[ \sum_{i=1}^{d} \left( \|\boldsymbol{g}_{1:T,i}\|_2 \right)^{2(1-pq)} \right]$$

where step $(a)$ uses the property that $a_j \leq a_{j-1}$ for all $j > 1$ and the results in Lemma 3, step $(b)$ uses $pq < 1$, $p, q > 0$, and step $(c) - (d)$ follow from Holder's inequality. The proof is complete for (13).

## C  PROOF OF THEOREM 2

In brief, we use the technique of parameter transformation proposed in (Yang et al. (2016)) to study Game for solving stochastic nonconvex optimization. In particular, we let $\boldsymbol{x}_0 = \boldsymbol{x}_1$ and for each $t \geq 1$

$$\boldsymbol{z}_t = \boldsymbol{x}_t + \frac{\beta_1}{1 - \beta_1} (\boldsymbol{x}_t - \boldsymbol{x}_{t-1}) = \frac{1}{1 - \beta_1} \boldsymbol{x}_t - \frac{\beta_1}{1 - \beta_1} \boldsymbol{x}_{t-1}, \tag{24}$$

where $\boldsymbol{z}_1 = \boldsymbol{x}_1$ for the special case $t = 1$. We note that the technique has also been employed in (Zhou et al. (2018)) and (Chen et al. (2018)) for analyzing PAdam and a class of Adam-type methods. Instead of considering $\{\boldsymbol{x}_j\}_{j=0}^{T}$ directly, we tackle $\{\boldsymbol{z}_j\}_{j=1}^{T}$ as in the literature.

In (Zhou et al. (2018)) and (Chen et al. (2018)), the authors provided a general upper bound for $f(\boldsymbol{z}_{t+1}) - f(\boldsymbol{z}_t)$, which also holds for Game. We now summarize their result in a lemma below:

**Lemma 4.** *Suppose the sequence $\{z_j\}_{j=1}^T$ is as defined in (24). Then under Assumption 1, the quantity $f(z_{t+1}) - f(z_t)$ is upper bounded by*

$$f(z_2) - f(z_1) \leq -\nabla f(x_1)^T \alpha_1 \hat{V}_1^{-p} g_1 + 2L \|\alpha_1 \hat{V}_1^{-p} g_1\|_2^2 \tag{25}$$

$$f(z_{t+1}) - f(z_t) \leq -\nabla f(x_t)^T \alpha_{t-1} \hat{V}_{t-1}^{-p} g_t + \frac{1}{1 - \beta_1} G_\infty^2 (\|\alpha_{t-1} \hat{v}_{t-1}^{-p}\|_1 - \|\alpha_t \hat{v}_t^{-p}\|_1)$$

$$+ 2L \|\alpha_t \hat{V}_t^{-p} g_t\|_2^2 + 4L \left( \frac{\beta_1}{1 - \beta_1} \right)^2 \|x_t - x_{t-1}\|_2^2 \quad t \geq 2. \tag{26}$$

With Lemma 4, we are ready to present the proof for Theorem 2. Firstly, taking expectation on both sides of (25) produces

$$\mathbb{E}[f(z_2) - f(z_1)] \leq \mathbb{E} \left[ -\nabla f(x_1)^T \alpha_1 \hat{V}_1^{-p} g_1 + 2L \|\alpha_1 \hat{V}_1^{-p} g_1\|_2^2 \right]$$

$$\leq \mathbb{E} \left[ d\alpha_1 \|\nabla f(x_1)\|_\infty \cdot \|\hat{V}_1^{-p} g_1\|_\infty + 2L \|\alpha_1 \hat{V}_1^{-p} g_1\|_2^2 \right]$$

$$\overset{(a)}{\leq} \mathbb{E} \left[ \frac{d\alpha_1}{(1 - \beta_2)^p} \|\nabla f(x_1)\|_\infty \cdot \||g|^{1-pq}\|_\infty + 2L \|\alpha_1 \hat{V}_1^{-p} g_1\|_2^2 \right]$$

$$\overset{(b)}{\leq} \mathbb{E} \left[ \frac{d\alpha_1 G_\infty^{2-pq}}{(1 - \beta_2)^p} + 2L \|\alpha_1 \hat{V}_1^{-p} g_1\|_2^2 \right], \tag{27}$$

where step $(a)$ uses $\hat{V}_1 = (1 - \beta_2)|g_1|^q$ and $pq < 1$, and step $(b)$ uses $\|\nabla f_t(x; \xi_t)\|_\infty \leq G_\infty$.

Next by rearranging terms and taking expectation in (26), we have

$$\mathbb{E} \left[ f(z_{t+1}) + \frac{G_\infty^2}{1 - \beta_1} \|\alpha_t \hat{v}_t^{-p}\|_1 - \left( f(z_t) + \frac{G_\infty^2}{1 - \beta_1} \|\alpha_{t-1} \hat{v}_{t-1}^{-p}\|_1 \right) \right]$$

$$\leq \mathbb{E} \left[ -\nabla f(x_t)^t \alpha_{t-1} \hat{V}_{t-1}^{-p} g_t + 2L \|\alpha_t \hat{V}_t^{-p} g_t\|_2^2 + 4L \left( \frac{\beta_1}{1 - \beta_1} \right)^2 \|x_{t-1} - x_t\|_2^2 \right]$$

$$\overset{(a)}{=} -\nabla f(x_t)^t \alpha_{t-1} \hat{V}_{t-1}^{-p} \nabla f(x_t) + \mathbb{E} \left[ 2L \|\alpha_t \hat{V}_t^{-p} g_t\|_2^2 + 4L \left( \frac{\beta_1}{1 - \beta_1} \right)^2 \|\alpha_{t-1} \hat{V}_{t-1}^{-p} m_{t-1}\|_2^2 \right]$$

$$\overset{(b)}{\leq} -\alpha_{t-1} \|\nabla f(x_t)\|_2^2 (G_\infty^{pq})^{-1} + \mathbb{E} \left[ 2L \|\alpha_t \hat{V}_t^{-p} g_t\|_2^2 + 4L \left( \frac{\beta_1}{1 - \beta_1} \right)^2 \|\alpha_{t-1} \hat{V}_{t-1}^{-p} m_{t-1}\|_2^2 \right], \tag{28}$$

where step $(a)$ uses the property that $\mathbb{E}[g_t] = \nabla f(x_t)$, and step $(b)$ uses the inequality $\|\hat{v}_{t-1}\|_\infty \leq G_\infty^q$ which can be derived from $\|\nabla f_t(x; \xi_t)\|_\infty \leq G_\infty, t \geq 1$.

Taking the summation of (27)-(28) over $t = 1$ to $T$ produces

$$(G_\infty^{pq})^{-1} \sum_{t=2}^T \alpha_{t-1} \mathbb{E} \|\nabla f(x_t)\|_2^2$$

$$\leq \mathbb{E} \left[ f(z_1) + \frac{G_\infty^2 \|\alpha_1 \hat{v}_1^{-p}\|_1}{1 - \beta_1} + \frac{d\alpha_1 G_\infty^{2-pq}}{(1 - \beta_2)^p} - \left( f(z_{T+1}) + \frac{G_\infty^2}{1 - \beta_1} \|\alpha_T \hat{v}_T^{-p}\|_1 \right) \right]$$

$$+ 2L \sum_{t=1}^T \mathbb{E} \left[ \|\alpha_t \hat{V}_t^{-p} g_t\|_2^2 \right] + 4L \left( \frac{\beta_1}{1 - \beta_1} \right)^2 \sum_{t=2}^T \mathbb{E} \left[ \|\alpha_{t-1} \hat{V}_{t-1}^{-p} m_{t-1}\|_2^2 \right]$$

$$\overset{(a)}{\leq} \mathbb{E} \left[ \Delta f + \frac{G_\infty^2 \|\alpha_1 \hat{v}_1^{-p}\|_1}{1 - \beta_1} + \frac{d\alpha_1 G_\infty^{2-pq}}{(1 - \beta_2)^p} \right] + 2L \sum_{t=1}^T \mathbb{E} \left[ \|\alpha_t \hat{V}_t^{-p} g_t\|_2^2 \right]$$

$$+ 4L \left( \frac{\beta_1}{1 - \beta_1} \right)^2 \sum_{t=1}^T \mathbb{E} \left[ \|\alpha_t \hat{V}_t^{-p} m_t\|_2^2 \right], \tag{29}$$

where step $(a)$ uses the inequality $\Delta f = f(x_1) - \inf_x f(x) \geq f(x_1) - f(z_{T+1})$ and $z_1 = x_1$.

Finally, substituting (12)-(13) into (29) produces

$$\mathbb{E}\|\nabla f(\boldsymbol{x}_{out})\|_2^2$$

$$= \frac{1}{\sum_{t=2}^T \alpha_{t-1}} \sum_{t=2}^T \alpha_{t-1}\mathbb{E}\|\nabla f(\boldsymbol{x}_t)\|_2^2$$

$$\leq \frac{G_\infty^{pq}}{\sum_{t=2}^T \alpha_{t-1}}\mathbb{E}\left[\Delta f + \frac{G_\infty^2\|\alpha_1\hat{\boldsymbol{v}}_1^{-p}\|_1}{1-\beta_1} + \frac{d\alpha_1 G_\infty^{2-pq}}{(1-\beta_2)^p}\right]$$

$$+ \frac{2LG_\infty^{pq}}{\sum_{t=2}^T \alpha_{t-1}}\frac{\alpha_1^2 T^{pq}}{(1-\beta_2)^{2p}}\mathbb{E}\left(\sum_{i=1}^d (\|\boldsymbol{g}_{1:T,i}\|_2)^{2(1-pq)}\right)$$

$$+ \frac{4LG_\infty^{pq}}{\sum_{t=2}^T \alpha_{t-1}}\left(\frac{\beta_1}{1-\beta_1}\right)^2 \frac{\alpha_1^2 G_\infty^{(1+r-2pq)}T^{(1+r)/2}}{(1-\beta_2)^{2p}}\mathbb{E}\left(\sum_{i=1}^d (\|\boldsymbol{g}_{1:T,i}\|_2)^{1-r}\right)$$

$$\overset{(a)}{=} \frac{G_\infty^{pq}\Delta f}{(T-1)\alpha_1} + \frac{1}{T-1}\left(\frac{G_\infty^{2+pq}\|\hat{\boldsymbol{v}}_1^{-p}\|_1}{1-\beta_1} + \frac{dG_\infty^2}{(1-\beta_2)^p}\right)$$

$$+ \frac{T^{pq}}{T-1} \cdot \frac{2LG_\infty^{pq}\alpha_1}{(1-\beta_2)^{2p}}\mathbb{E}\left(\sum_{i=1}^d (\|\boldsymbol{g}_{1:T,i}\|_2)^{2(1-pq)}\right)$$

$$+ \frac{T^{(1+r)/2}}{T-1} \cdot \frac{4LG_\infty^{1+r-pq}\alpha_1}{(1-\beta_2)^{2p}}\left(\frac{\beta_1}{1-\beta_1}\right)^2 \mathbb{E}\left(\sum_{i=1}^d (\|\boldsymbol{g}_{1:T,i}\|_2)^{1-r}\right)$$

$$= \frac{M_1}{T-1} + \frac{M_2 d}{T-1} + \frac{M_3 T^{pq}}{T-1}\mathbb{E}\left(\sum_{i=1}^d (\|\boldsymbol{g}_{1:T,i}\|_2)^{2(1-pq)}\right)$$

$$+ \frac{M_4 T^{(1+r)/2}}{T-1}\mathbb{E}\left(\sum_{i=1}^d (\|\boldsymbol{g}_{1:T,i}\|_2)^{1-r}\right),$$

where step $(a)$ uses the property $\alpha_t = \alpha_1$ for all $t \geq 2$, and $\{M_l\}_{l=1}^4$ are given by (15)-(18). The proof is complete.

## D  CNN ARCHITECTURES IN EXPERIMENTS

**Remark 3.** *The full name of SERLU in Table 3 is so called* scaled exponentially-regularized linear unit. *Furthermore, the shift-dropout in Table 3 is designed specifically for SERLU, which includes dropout as a special case.*

Table 2: CNN for MNIST using SERLU

| Layer 1 | conv.: $3 \times 3$@32  (SERLU) |
|---------|-------------------------------|
| Layer 2 | conv.: $3 \times 3$@64 (SERLU) 
 max-pooling |
| Layer 3 | dense: 512 neurons  (SERLU) |
| Layer 4 | dense + softmax |

Table 3:  CNN for CIFAR10 using SERLU

| Layer 1 | conv.: $3 \times 3$@32  (SERLU) |
|---------|-------------------------------------------------|
| Layer 2 | conv.: $3 \times 3$@32 (SERLU)
max-pooling
shift-dropout |
| Layer 3 | conv.: $3 \times 3$@64  (SERLU) |
| Layer 4 | conv.: $3 \times 3$@64 (SERLU)
max-pooling
shift-dropout |
| Layer 5 | conv.: $3 \times 3$@128  (SERLU) |
| Layer 6 | conv.: $3 \times 3$@128 (SERLU)
max-pooling
shift-dropout |
| Layer 7 | dense: 512 neurons (SERLU)
shift-dropout |
| Layer 8 | dense + softmax |

Table 4:  CNN for CIFAR10 using ELU

| Layer 1 | conv.: $3 \times 3$@32  (ELU) |
|---------|-------------------------------------------------|
| Layer 2 | conv.: $3 \times 3$@32 (ELU)
max-pooling
dropout |
| Layer 3 | conv.: $3 \times 3$@64  (ELU) |
| Layer 4 | conv.: $3 \times 3$@64 (ELU)
max-pooling
dropout |
| Layer 5 | conv.: $3 \times 3$@128  (ELU) |
| Layer 6 | conv.: $3 \times 3$@128 (ELU)
max-pooling
dropout |
| Layer 7 | dense: 512 neurons (ELU)
dropout |
| Layer 8 | dense + softmax |

Table 5:  CNN for CIFAR10 using ReLU

| Layer 1 | conv.: $3 \times 3$@32 (ReLU) |
|---------|-------------------------------------------------|
| Layer 2 | conv.: $3 \times 3$@32 (ReLU)
max-pooling
dropout |
| Layer 3 | conv.: $3 \times 3$@64  (ReLU) |
| Layer 4 | conv.: $3 \times 3$@64 (ReLU)
max-pooling
dropout |
| Layer 5 | conv.: $3 \times 3$@128  (ReLU) |
| Layer 6 | conv.: $3 \times 3$@128 (ReLU)
max-pooling
dropout |
| Layer 7 | dense: 512 neurons (ReLU)
dropout |
| Layer 8 | dense + softmax |

