# OpenReview forum: "GENERALIZED ADAPTIVE MOMENT ESTIMATION"
_ICLR.cc/2019/Conference_

### Official Review · AnonReviewer2 · 2018-10-31
**acceptable**

**Rating:** 7
**Confidence:** 4

**Review:**

The authors proposed a generalized adaptive moment estimation method(Game). Compared to the existing methods AMSGrad and PAdam, the new method Game tracks only two parameters in iteration and hence saves memory. Besides, they introduced a additional tuning parameter $q$ to track the q-th moment of the gradient and allow more flexibility. The authors also provided the theoretical convergence analysis of Game for convex optimization and smooth nonconvex optimization. Their experiment shows Game may produce better performance than AMSGrad and PAdam with a little bit sacrifice of convergence speed. Game is a promising alternative method for training large-scale neural network.

---

### Official Review · AnonReviewer3 · 2018-11-02
**This paper proposes a new framework that generalizes previous algorithms such as AMSGrad and PAdam.**

**Rating:** 4
**Confidence:** 3

**Review:**

Pros:
1. The algorithm saves about 1/3 memory consumption compared with AMSGrad.
2. The authors give the proof that the generalized algorithms have the same convergence rate with weaker assumptions.

Cons:
1. All the experiments are based on CNN. There are no results based on modern deep neural networks such as Residual Nets and Dense Nets, where it is obvious to see Adam suffers from poor generalization.

2. Algorithms like SGD with momentum and Adam should be included for comparison.

3. This framework introduces two more hyper-parameters p and q, which makes it more difficult for practitioners to tune.

Although this framework has proven convergence in both convex and non-convex smooth cases, the experimental evidence is limited. In addition, the proof strategy is not novel enough, Theorem 1 is similar to Theorem 4 in AMSGrad paper and Theorem 2 is similar to Theorem 3.3 in Zhou et al's paper.

---

> ### Author Response · Authors · 2018-11-28
> **Response to AnonReviewer3**
>
> Many thanks for reading our paper and providing constructive comments.
>
> * Firstly, we have modified the new algorithm “Game” to include AMSGrad (by setting  p=0.5, q=2) and PAdam (by setting q=2)  as special cases. Experiments show that the new version of Game produces better results.
>
> * So far we have conducted experiments for using VGG-16 on CIFAR10 and CIFAR100, where the network employ the dropout and batch normalization techniques. We have evaluated five algrithms, which are Adam, AMSGrad, PAdam, SGD with momentum, and Game.   The results suggest that the new algorithm Game outperforms Adam, AMSGrad, and PAdam w.r.t. validation performance, and has comparable performance as SGD with moment.
>
> * Experiments on ResNet and wide resnet demonstrate that Game with (p,q) = (0.25, 4) still outperforms Adam and AMSGrad.
>
> *  We belive that the parameter p and q can be preset properly without tunning, which are similar to the parameters beta_1 and beta_2 of Adam. Our recommendation for p and q are p=0.25 and q = 4.  Therefore, the product pq=1, which is in line with that of AMSGrad (where p=0.5, q=2). We will conduct more experiments to confirm the effectiveness of the above setups.

---

### Official Review · AnonReviewer1 · 2018-11-02
**Approximative and weakly motivated Adam modification**

**Rating:** 3
**Confidence:** 4

**Review:**

Summary
------

The authors propose an adaptation of the Adam method, with the AMSGrad correction and an additional parameter to p to exponentiate the diagonal conditioning matrix V (Padam).

The proposed method changes two aspects: first, there is no need to retain two version of the rescaling matrix v, where amsgrad and Padam keeps the last monotone \hat v)t and non-monotone version v_t. Secondly, a new parameter q is introduced, that replaces the q=2 in the moment estimation phase of (P)Adam.

A regret analysis is proposed in the convex case, while a vanishing bound on the gradient is derived in the non-convex smooth case.

Review
------

Although improving optimization methods is certainly important for the machine learning community, the reviewer have strong concerns about this paper.

First of all, the paper is hard to read as it contains too many approximations. What does 'SGD is known to work reasonably well regardless of their problem structure' means ? Same thing for 'Its performance deteriorates when the gradients are dense due to a rapid decay of the learning rates.' The authors uses many times elliptical discourse to detail the course of their analysis, which is non informative: for instance, 'one can easily derive the upper bound expression', and 'It is not difficult to conclude that when G_t [...]'. This level of writing is not professional. Some completely irrelevant argument are proposed to justify the method: 'For instance the extension from l_2 norm to l_p norm and generalization from Cauchy-Schwart to Holder inequality'.

The reviewer has interrogations about the relevance of the proposed algorithm. The additional parameter q needs to be tuned, which carries only the promise of further overfitting. I would have been convinced by an sequence of experiment where q is set automatically by considering a validation set, and then tested on a left out test set. However, the authors report only the results for the best q, with non significant differences (and not quantified, there is no result tables). Using q=2 at least made sense from the point of view of empirical Fisher matrix approximation.

The review also have several concerns aout the correctness of the proposed arguments. First, the major argument of memory usage stems from 1) a miscalculation and 2) a misunderstanding of memory bottlenecks in deep learning. 1) adam models keeps in memory x_t (the model parameter), g_t (the model gradient), m_t (momentum) \hat v_t and v_{t-1} (the monotone and non monotone version of the second order moment estimation. In contrast, the proposed model do not track v_{t-1}: this amounts to a memory saving of 20% considering all model related parameters. 2) more importantly, the most important memory usage in deep leaning comes from the activations that need to be kept in memory during the forward pass to perform the backward pass. Even the biggest model are less than 1GB, and most of the memory used during training is dedicated to intermediary activations. This makes the major argument of the paper less convincing, and misleads the reader.

Second, even when disregarding the slightly abusive assumptions over the iterate sequences, that are common in the adaptive stochastic optimizers community, I think that the bound proposed in theorem 1 is non informative, as the second term behaves like T sqrt(T) assymptotically, due to the presence of 1 / \alpha_t. This does not show the convergence of averaged regret R_T / T.

Regarding the experiment section, I am afraid that testing a new optimizer over MNIST and CIFAR is not enough to show the relevance of the method for the whole deep learning community. An eperiment over a non-toy dataset (eg ImageNet), and on non computer-vision dataset (eg from NLP) would be a minimum, besides the overfitting concern described above.

In conclusion, it is the reviewer's opinion that significant rework in term of presentation and strong improvement of the experiment section to make the case for the Game optimizer.

Minor comments
------------
Table 1: what do you bound when you compare results ? I think there is a typo in Zhou et al. result: 1/2 should read p.

Eq (1): it is rather surprising to use x_t as the model parameters in the ICLR community.

p 7: the dimension d could be larger than T when training large-scale neural networks: how does it relate to comparing sqrt(dT) to (dT)^s ?

---

> ### Author Response · Authors · 2018-11-28
> **Response to AnonReviewer1**
>
> Many thanks for the providing constructive comments for our paper:
>
> * We have obtained comparison results regarding different setups of (p,q) from one experiment.  It is found that (q=0.25, p=4) performs better than other setups. The detailed comparisons will be reflected in revision of the paper. We are now conducting more experiments to further study the effectiveness of the setup (q=0.25, p=4).
>
> * In the revision, we have modified the new algorithm “Game” to include AMSGrad (by setting  p=0.5, q=2) and PAdam (by setting q=2)  as special cases.  So the memory usage of Game will be equivalent to AMSGrad and PAdam. Experiments show that the new version of Game produces better results.
>
> * Thanks for commenting the convergence analysis in Theorem 1. In the revision, we will discuss  the results of Theorem 1 in depth.
>
> * Regarding the experimental part, we are now evaluating VGG net and resnet, which will be included in the revision. If time allows, we will continue testing the method for other datasets.

---

### Public Comment · (anonymous) · 2018-11-04
**A paper literally combined results from our papers**

I am the author of two key papers (Zhou et al. (2018) https://arxiv.org/pdf/1808.05671.pdf  and Chen & Gu (2018) https://arxiv.org/pdf/1806.06763.pdf) cited in this submission. The way this paper is written makes me really upset.

This paper is heavily built up on our papers, and literally copied a huge part of the theorems and proofs from our paper (even the layout of the equations such as equation breaks, term decomposition is highly similar to our proofs). For example, the following sentences/paragraphs/notations are copied from our papers, i.e., Zhou et al. (2018) and Chen & Gu (2018):

1. In Theorem 2, the authors present their upper bound for \|\nabla f(x_{out})\|_2 by several constant parameters M_i. Compared with Theorem 3.3 in Zhou et al. (2018), the definition of M_i and the upper bound itself are highly similar to what Theorem 3.3 in Zhou et al. (2018) presents.

2. In the proof of Theorem 2, the use of notation is essentially the same as the proof of Theorem 3.3, section A.1 in Zhou et al. (2018). For instance, both of these two works set the output point as x_{out}, set the upper bound for \|\nabla f(x)\|_2 as G_\infty, etc.

3. The main proof roadmap for Theorem 2 is the same as the proof roadmap for Theorem 3.3 in Zhou et al. (2018). Both of these two work aim to bound \|\nabla f(x_t)\|_2^2 by the difference of H(t+1) and H(t), where H(t) = f(z_t) + \frac{G_\infty^2}{1-\beta_1}\|\alpha_{t-1}\hat{v}_{t-1}^{-p}\|_1. To get the bound, both of these two work provide upper bounds for \sum_{t=1}^T \alpha_t^2 \mathcal{E} \|\hat{V}_t^{-p} m_t\|_2^2 and \sum_{t=1}^T \alpha_t^2 \mathcal{E} \|\hat{V}_t^{-p} g_t\|_2^2. However, the authors did not point out this part of proof is essentially a twist of our proof.

4. In the proof of Theorem 2, the equation (27) at page 13 is essentially the same as the equation (A.13) at page 13 in Zhou et al. (2018), without appropriate reference.

5. In the proof of Theorem 2, the equation (28) at page 13 is essentially the same as the equation (A.14) at page 14 in Zhou et al. (2018), without appropriate reference.

6. In the proof of Theorem 2, the equation (29) at page 13 is essentially the same as the equation (A.15) at page 14 in Zhou et al. (2018), without appropriate reference.

---

> ### Author Response · Authors · 2018-11-08
> **Response to the author of PAdam**
>
> Many thanks for reading our paper and providing comments.
>
> Firstly in the introduction, we claim two contributions. The first one is to introduce an additional parameter q in your algorithm PAdam, which is referred to as Game in this paper. Experiments show that the parameter q plays a really important role to improve the generalization performance of AMSGrad, which will be reflected in the revision.
>
> Our 2nd contribution is about removing the constraint of beta_1 and beta_2 in theoretical analysis, which is NEW to deep learning community to our best knowledge. In the main context of Section 4, we emphasize the 2nd contribution rather than a new convergence analysis approach.
>
> We indeed followed one of your two papers on nonconvex convergence analysis, which is put in the appendix for the readers to understand the derivation procedure. In the revision, we will carefully reshape the proof for Theorem 2 in the appendix to point out the original innovation of your research work.

---

> > ### Author Response · Authors · 2018-11-28
> > **On the importance of the additional parameter q in the new training method**
> >
> > Our recent experiments on VGG-16 over CIFAR10 strongly suggest that it is very important to introduce a new parameter q in your algorithm PAdam, which is referred to as Game in the current paper.  It is found that if a small p is chosen in PAdam, then a large q value is preferred to have a balance w.r.t. the individual learning rates.  The experiments suggest that p=0.25 and q=4 is a good setup. We are now conducting more experiments to confirm the effetiveness of the setup.
> >
> > The key message from our research is that both p and q parameters shouldn’t be set small at the same time from the aspect of generalization performance, even though small values of p and q making the algorithm more close to SGD with momentum.  We recommend to set pq=1. Our experiments so far suggest that the setup p=0.25 and q = 4 is a good candidate.  We hope our above observation could benefit the deep learning community.

---

### Author Response · Authors · 2018-12-03
**Main points of the revised paper**

We have finished refining our paper based on all the comments of the reviewers and the author of PAdam.

* Firstly, we notice that the new version of Game includes AMSGrad and PAdam as special cases by setting (p,q) = (0.5, 2) and q = 2, respectively. The introduction of the additional parameter q in Game is for balancing the effect of parameter p when using the information of gradient magnitude for training DNNs.  When a small parameter p is selected, one can consider the option of choosing a large parameter q to make the individual learning rates in a reasonable range. Conceptually speaking, the extension from AMSGrad to Game is similar to the generalization from l2 norm to lp norm.

* In the revision, we have compared five methods:  SGD with moment, Adam, AMSGrad, PAdam, and Game for training three different neural networks over CIFAR10 and CIFAR100, which are VGG-16, ResNet-18, and wide ResNet. It is found that when the (p,q) of Game is set to be (0.25, 4), Game outperforms Adam and AMSGrad for all the three neural networks w.r.t. the validation performance. For the case of VGG-16,  the performance of Game is comparable to that of SGD with moment w.r.t. the validation peformance.

* Based on the comments of the authors of PAdam, the theoretical analysis is reshaped in the revision to differentiate our contribution from theirs. To briefly summarize, it is found in the current paper that, in principle, the parameter beta_2 can be chosen freely without considering beta_1 while traditional theoretical analysis for AMSGrad and PAdam requires that beta_1 < \sqrt{beta_2}. We hope our new findings  can bring insights to develop more advanced training methods in the future.

---

### Meta-Review · Area_Chair1 · 2018-12-15
**Lacking in quality, clarity and questionable correctness and originality**

**Confidence:** 5
**Recommendation:** Reject

**Metareview:**

The reviewers find the per difficult to read. Reviewers also had concerns regarding the correctness of various claims in the paper. The paper was also found lacking in experimental analysis, as it only tested on relatively small datasets, and only no a CNN architecture. Overall, the paper appears to be lacking in quality and clarity, and questionable in correctness and originality.